# Unrolled Networks are Conditional Probability Flows in MRI Reconstruction

## Abstract

Magnetic Resonance Imaging (MRI) offers excellent soft-tissue contrast without ionizing radiation, but its long acquisition time limits clinical utility. Recent methods accelerate MRI by under-sampling $k$-space and reconstructing the resulting images using deep learning. Unrolled networks have been widely used for the reconstruction task due to their efficiency, but suffer from unstable evolving caused by freely-learnable parameters in intermediate steps. In contrast, diffusion models based on stochastic differential equations offer theoretical stability in both medical and natural image tasks but are computationally expensive. In this work, we introduce flow ODEs to MRI reconstruction by theoretically proving that unrolled networks are discrete implementations of conditional probability flow ODEs. This connection provides explicit formulations for parameters and clarifies how intermediate states should evolve. Building on this insight, we propose Flow-Aligned Training (FLAT), which derives unrolled parameters from the ODE discretization and aligns intermediate reconstructions with the ideal ODE trajectory to improve stability and convergence. Experiments on three MRI datasets show that FLAT achieves high-quality reconstructions with up to $3\times$ fewer iterations than diffusion-based generative models and significantly greater stability than unrolled networks.

## 1 Introduction

Magnetic Resonance Imaging (MRI) is widely used in clinical diagnosis (Mahlknecht et al., 2010; Rocca et al., 2024) and biomedical research (Scarciglia et al., 2025) due to its high soft-tissue contrast and lack of ionizing radiation. It samples acquisitions in the frequency domain (also called $k$-space) and then converts them to the image domain using inverse Fourier Transform (iFT) (Plewes & Kucharczyk, 2012). However, sampling the entire $k$-space requires long acquisition time, raising the risk of motion artifacts, increasing patient discomfort, and lowering clinical throughput (Lustig et al., 2008). Accelerated MRI techniques, such as Compressed Sensing (CS) (Donoho, 2006; Lustig et al., 2008), reduce acquisition time by under-sampling in the $k$-space domain. However, recovering the high-quality images from these under-sampled measurements is an ill-posed problem. See Fig. 1 I) for an illustration. Thus, one-step reconstruction is difficult in this ill-posed setting, making it natural to adopt iterative approaches (Block et al., 2009) that progressively refine a reconstruction.

The emergence of deep learning has enabled powerful iterative schemes. Unrolled networks (Sun et al., 2016; Aggarwal et al., 2018; Zhang & Ghanem, 2018; Sriram et al., 2020; Schlemper et al., 2017; Aghabiglou & Eksioglu, 2021) have become one of the most successful approaches for MRI reconstruction. It consists of a series of sub-networks (cascades), where each cascade corresponds to an unrolled iteration of a classical algorithm such as first-order optimization method (Zhang & Ghanem, 2018) or ADMM (Sun et al., 2016); each iteration moves the estimate one step closer to the final result. By solving a sequence of smaller reconstruction subproblems rather than attempting complete recovery in a single step, unrolled networks achieve promising reconstruction quality.

However, despite their widespread adoption, unrolled networks suffer from fundamental limitations. First, they are typically trained with supervision only at the final cascade. This creates the issue of diminishing gradients across the sub-networks/cascades, with earlier cascades receiving weak or noisy gradient signals and thus failing to learn any meaningful updates. In this paradigm, only the last cascade truly learns how to denoise and recover details, resulting in under-usage of the

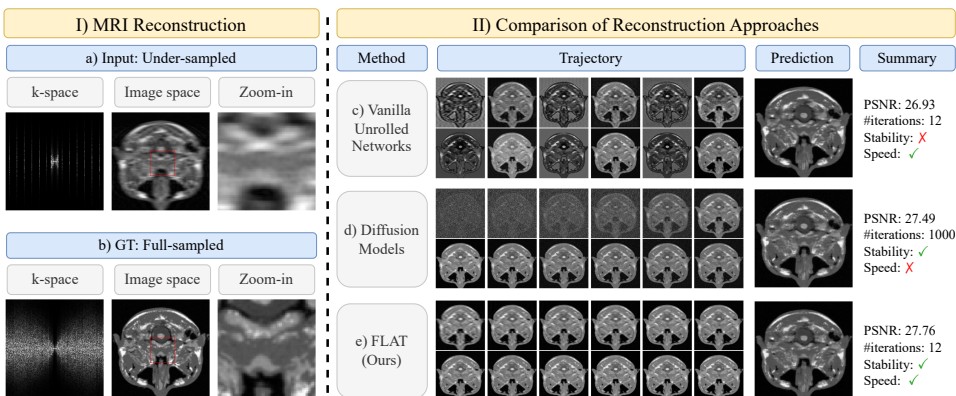

Figure 1: (I) Illustrating the MRI reconstruction task: from an under-sampled, aliased input (a), the task is to recover the clean, fully-sampled image (b). (II) Comparison of reconstruction approaches: c) Vanilla unrolled networks achieve fast results with few iterations but suffer from unstable intermediate steps that degrade image quality. d) Diffusion models achieve high image quality but require many iterations, adversely impacting speed. e) Our FLAT, grounded in probability flow ODEs, achieves high reconstruction quality with a low iteration count compared with diffusion-based generative models, yielding both stability and speed.

early cascades. Though some unrolled networks use intermediate supervision, they often lack a generative model perspective and thereby limits the performance. Second, parameters such as step size and weighting coefficients are typically learned in the training process. Without theoretical guidance, they lack a holistic control of all cascades to best coordinate their efforts. Fig. 1 II) c) illustrates these issues: the sequence of intermediate outputs shows erratic oscillatory behavior, with each image differing significantly from the previous one, rather than smooth convergence toward the final reconstruction.

Differential equation-based generative modeling has also inspired new MRI reconstruction strategies. Diffusion models such as DDPM (Ho et al., 2020) and DDIM (Song et al., 2020a), based on Stochastic Differential Equations (SDE), progressively denoise data within a probabilistic framework and have achieved state-of-the-art results in MRI reconstruction (Xie & Li, 2022; Cao et al., 2024). Despite their strong performance, they require at least 50 iterative steps to achieve acceptable image quality, with best results at 1000 or more steps (see Fig. 1 II) d)). This makes them computationally expensive, severely limiting their practical utility. This motivates the need for an approach that can achieve strong quality in minimal steps, without sacrificing one for the other. In computer vision, a promising direction is Ordinary Differential Equations (ODE) flow-based generation, which describes data generation as continuous probability transport (Lipman et al., 2022; Chen et al., 2023; Song et al., 2020b; Albergo & Vanden-Eijnden, 2022). Recent flow-based works (Chen et al., 2025; Bu et al., 2025; Gu et al., 2025; Qin et al., 2025) have demonstrated efficient and stable sampling in natural image synthesis, guaranteeing low iteration counts while maintaining quality. Despite these advantages, flow ODEs remain unexplored in MRI reconstruction.

This gap presents a compelling opportunity: flow ODEs have proven advantages in efficiency and stability for image generation, yet lack specialized formulations for MRI reconstruction. Meanwhile, unrolled networks dominate MRI reconstruction but suffer from fundamental training instabilities and lack of generative model perspective. We uncover a key insight that these seemingly different approaches are more connected than previously understood, enabling us to leverage the theoretical rigor of flow ODEs to address the limitations of unrolled networks in MRI reconstruction.

In this work, we introduce flow ODEs to MRI reconstruction by theoretically proving that unrolled networks are discrete implementations of energy score based conditional probability flow ODEs. To the best of our knowledge, this is the first work to prove this fundamental connection, allowing us to bring the proven advantages of flow ODEs to MRI reconstruction for the first time. The key idea is that MRI reconstruction can be viewed as a trajectory from an under-sampled image to a high-quality image: unrolled networks take discrete "steps" along this path, while flow ODEs describe the same path as a continuous trajectory. Our mapping reveals that each cascade step corresponds to

a specific time point along the ODE path, and crucially, that the seemingly heuristic parameters in unrolled networks (like step sizes) should follow ODE discretization rules.

From this fundamental connection, we have three critical implications: (i) cascade timesteps must satisfy specific constraints to ensure complete trajectory traversal, (ii) parameters have explicit formulations grounded in ODE theory rather than being free, and (iii) intermediate supervision can be applied using ODE-consistent targets to align unrolled networks with ideal trajectories. Building on this theory, we propose Flow-Aligned Training (FLAT), a training framework that constrains unrolled network parameters based on ODE discretization, and provides intermediate supervision along optimal ODE paths. By grounding unrolled network training in ODE theory, FLAT improves intermediate steps' stability, enables better control over iterative updates, and still benefits from the computational efficiency of unrolled networks. Fig. 1 II) e) shows that FLAT achieves high reconstruction quality with low iteration counts and stable intermediate outputs.

We evaluate FLAT on three public MRI datasets: Brainweb (Cocosco et al., 1997), MR-BrainS13 (Mendrik et al., 2024), and fastMRI (Zbontar et al., 2018). Experiments show that FLAT successfully bridges flow ODEs and MRI reconstruction and outperforms existing methods, achieving superior reconstruction quality with up to $3\times$ fewer iterative steps than diffusion-based methods and significantly enhanced stability compared to vanilla unrolled networks. This demonstrates that our theoretical framework delivers practical advantages over both traditional unrolled approaches and state-of-the-art diffusion models. In summary, our contributions are as follows:

- To the best of our knowledge, we are the first to introduce flow ordinary differential equations (ODEs) to the MRI reconstruction task by theoretically proving that unrolled networks are discrete implementations of conditional probability flow ODE, establishing a fundamental equivalence between these two paradigms.

- We propose Flow-Aligned Training (FLAT), which enforces an ODE-consistent cascade schedule, grounds previously free hyperparameters (such as step sizes and weighting terms) in the ODE view, and adds intermediate supervision to align the network's updates with the ideal ODE trajectory, thereby improving stability and interpretability.

- Extensive experiments demonstrate that FLAT improves reconstruction performance of unrolled networks on multiple MRI benchmarks, achieving state-of-the-art results with fast and stable inference.

## 2 RELATED WORK

**Deep Learning-based MRI Reconstruction.** Inspired by iterative optimization algorithms, unrolled networks such as ADMM-Net (Sun et al., 2016), MoDL (Aggarwal et al., 2018), Cascaded U-Net (Aghabiglou & Eksioglu, 2021) and E2E-VarNet (Sriram et al., 2020) unfold iterative solvers into trainable cascades that interleave learned regularization and data consistency. Transformer-based architectures (Huang et al., 2022; Guo et al., 2023; Zhou et al., 2023) have been introduced to better capture long-range dependencies across image and $k$-space domains. Recently, state-space models (SSMs) such as Mamba have been adapted to MRI reconstruction to combine long-range context modeling with linear-time complexity (Korkmaz & Patel, 2025; Meng et al., 2025; Ji et al., 2024; Zou et al., 2025; Joo et al., 2025). Finally, diffusion models for accelerated MRI (Xie & Li, 2022; Cao et al., 2024; Güngör et al., 2023) established the stochastic differential equation (SDE) plus data consistency paradigm.

**Flow ODE for Image Synthesis.** Probability Flow ODEs links reverse SDE sampling and ODE transport (Song et al., 2020b). Earlier works (Liu et al., 2022; Albergo & Vanden-Eijnden, 2022; Tong et al., 2023) train Continuous Normalizing Flows (CNFs) to learn maps between two data distributions. Recent works such as PixelFlow (Chen et al., 2025), HiFlow (Bu et al., 2025), STARFlow (Gu et al., 2025) and ResFlow (Qin et al., 2025) focus on image synthesis in specific domains. Additionally, Yazdani et al. (2025) introduces flow matching in medical image synthesis, utilizing flow-based training for faster and higher-quality medical image generation. However, this approach synthesizes images, and is unable to reconstruct images from under-sampled $k$-space data.

## 3 METHOD

We aim to introduce flow ODEs to the MRI reconstruction task by establishing the theoretical equivalence between unrolled networks and energy based conditional probability flow ODEs. We first introduce the necessary background on MRI reconstruction, unrolled networks, and flow ODEs in Sec. 3.1. We then present our proof that unrolled networks can be interpreted as discrete implementations of energy based conditional probability flow ODEs, where each cascade corresponds to a discrete timestep along the continuous ODE trajectory in Sec. 3.2. In Sec. 3.3, we reveal that this equivalence has three critical implications: ❶ the cascade timesteps must satisfy specific constraints to ensure complete traversal of the reconstruction trajectory, ❷ parameters like step sizes have formulations grounded in ODE theory rather than being free parameters, and ❸ intermediate supervision can be applied using ODE-consistent targets to align the unrolled network with the ideal trajectory. Finally, we introduce our Flow-Aligned Training (FLAT) framework in Sec. 3.4, which builds upon the implications to design training strategies for stable and efficient MRI reconstruction.

### 3.1 PRELIMINARIES

**MRI Sampling.** The compressed sensing (CS)-based MRI acquisition process can be expressed as

$$y = Ax + \epsilon \tag{1}$$

where $x$ denotes the 2-D original image in $k$-space, $y$ represents the 2-D observed (under-sampled) $k$-space image, $A$ is the sampling matrix which is known a priori in the MRI reconstruction task, and $\epsilon \sim \mathcal{N}(\mu_e, \Sigma_e^2)$ is Gaussian noise. For simplicity, we assume $\epsilon \sim \mathcal{N}(0, \sigma^2)$. In conventional CS-based MRI reconstruction, the goal is to recover $x$ by solving the following optimization problem:

$$\hat{x} = \arg\min_x \frac{1}{2}\|Ax - y\|^2 + \nu\Psi(x) \tag{2}$$

where $\hat{x}$ is the estimated clean image in $k$-space, $\Psi(x)$ denotes a sparsity-inducing regularization term, and $\nu$ is the hyperparameter that controls the level of regularization. We claim that all these variables, including $x, y, A, \epsilon$, are in $k$-space. Though the convolutional neural networks process in image space, the $\Psi(x)$ term takes $k$-space data as input, converts to image space, processed by CNN, and converts back to $k$-space, so the term is still in $k$-space.

**Unrolled MRI Reconstruction.** Unrolled networks are well-equipped to address the optimization problem in Eq. (2). In this work, we focus on unrolled networks that utilizes first-order gradients in optimization. We consider E2E-VarNet as our base unrolled network, which unrolls first-order optimization method into $K$ cascades or iterations. Let $x^{(K)} = A^T y$ denote the initial reconstruction (i.e., the under-sampled observation) and $x^{(0)}$ the final reconstruction result after $K$ iterations. Then, the $k^{th}$ iteration is formulated as:

$$x^{(k-1)} = x^{(k)} - \underbrace{\eta_k A^T(Ax^{(k)} - y)}_{data\ consistency} + \underbrace{\eta_k \mu \Phi_k(x^{(k)})}_{regularization} \tag{3}$$

where $x^{(k)}$ is the network output at iteration $k$ (with $k$ decreasing from $K$ to 0), $\eta_k$ is the learnable step size, $\Phi_k(\cdot)$ is proximal regularization block implemented with CNN. In E2E-VarNet, training is end-to-end with supervision applied only to the final cascade, i.e., no explicit supervision on intermediate outputs which can leave them under-constrained.

**Flow ODE Based Image Synthesis.** Flow-based ordinary differential equations (ODEs) provide a continuous pathway that smoothly transitions from one image distribution to another. To be consistent with diffusion models, we use index 1 for the source images, and index 0 for the clean images. Consider two image distributions $\pi_1$ (the source distribution, typically noise or low-resolution images) and $\pi_0$ (the target distribution, real or high-quality images). Flow-based ODE image synthesis methods learn a time-dependent vector field $v$ that can be used to construct a time-dependent path (called *flow*) to transport samples from $\pi_1$ to $\pi_0$. Let $\{x_t\}_{t\in[0,1]}$ denote the path of a sample under this flow, defined by the ordinary differential equation $\frac{dx_t}{dt} = v(x_t, t)$ with $x_0 \sim \pi_0$ and $x_1 \sim \pi_1$. In practice, $v$ is learnt using a neural network $v_\theta$, and trained so that its trajectory aligns with a simple, linearly parameterized path from the source to the target. To this end, several works (Liu et al., 2022;

Chen et al., 2025; Yazdani et al., 2025) supervise $v_\theta$ against the constant straight-line interpolation of velocity:

$$x_t = tx_1 + (1-t)x_0 \implies \frac{dx_t}{dt} = x_1 - x_0, \tag{4}$$

by minimizing a time-averaged least-squares objective: $\min_\theta \int_0^1 \mathbb{E}\left[\|(x_1 - x_0) - v_\theta(x_t, t)\|_2^2\right] dt$. This objective encourages the learned vector field to point along the linear direction from $x_1$ toward $x_0$ at every intermediate state $x_t$. Intuitively, it plays a role akin to the step-by-step diffusion process, except the evolution is deterministic and governed by an ODE.

### 3.2 A Conditional Probability Flow Perspective of Unrolled Networks for MRI Reconstruction

We now state and prove our main equivalence result between unrolled networks and energy based conditional probability flow ODEs for MRI reconstruction.

**Theorem 1.** *Each cascade of a first-order unrolled network can be viewed as one discrete step along ODE's continuous trajectory, i.e., an unrolled network is a time-discretization (e.g., forward Euler) of an energy-based conditional probability flow ODE.*

*Proof.* We formalize the proof as follows. Let the MRI reconstruction task be modeled using a conditional flow ODE evolving from the under-sampled initialization $x_1 = A^T y$ towards the fully-sampled $x_0$:

$$\frac{dx_t}{dt} = v(x_t, t; y) \tag{5}$$

where $v(x_t, t; y)$ denotes the velocity field at intermediate position $x$, timestep $t$ conditioned on the observation $y$, and $\lambda(t)$ is a time-dependent scaling function. To achieve a high energy score for $x$ close to $x_1$, and low score for $x$ close to $x_0$, one can build up an energy based model upon $x$ and $y$:

$$E(x; y) = \frac{1}{2\sigma^2}\|Ax - y\|^2 - \log p(x) \tag{6}$$

Here $\sigma$ is a normalized scale factor. The first term indicates data consistency between $Ax$ and $y$, and the second term is a log density of prior. Then, we define a pseudo posterior by setting the negative log-density equal to this energy

$$p_\phi(x|y) \propto \exp(-E_\phi(x; y)) \tag{7}$$

where $p_\phi(x|y)$ is a pseudo posterior that approximates the true posterior $p(x|y)$, and $p_\phi(x)$ is the learnable prior. Our conditional probability flow ODE is then defined on this energy-based pseudo conditional probability density function:

$$v(x_t, t; y) = \lambda(t)\nabla_x \log p_\phi(x_t|y) \tag{8}$$

Decompose Eq. (8) with Eq. (6) and Eq. (7), and represent the learnable term $\nabla_x \log p_\phi(x_t)$ with a neural network $v_\theta(x_t, t)$ gives

$$\frac{dx_t}{dt} = \lambda(t)v_\theta(x_t, t) - \lambda(t)A^T(Ax_t - y)/\sigma^2 \tag{9}$$

To solve this continuous ODE, we apply numerical discretization. Using a forward Euler step from $t_k$ to $t_{k-1}$ with $\delta_k = t_{k-1} - t_k$ (ignoring the $o(\delta_k^2)$ term[1]):

$$x_{t_{k-1}} = x_{t_k} + \underbrace{\delta_k \lambda(t_k)v_\theta(x_{t_k}, t_k)}_{pseudo\ prior} - \underbrace{\delta_k \lambda(t_k)A^T(Ax_{t_k} - y)/\sigma^2}_{data\ consistency} \tag{10}$$

For clarity, we color-code terms from discretized conditional probability flow ODE and unrolled network iteration. Notice how Eq. (10)'s *data consistency* and *pseudo prior* terms are equivalent to the *data consistency* and *regularization* terms from Eq. (3). We thus establish the following correspondence:

$$x^{(k)} = x_{t_k}, \qquad \eta_k = \delta_k \lambda(t_k)/\sigma^2, \qquad \mu = -\sigma^2, \qquad \Phi_k(x_k) = v_\theta(x_{t_k}, t_k) \tag{11}$$

$\square$

---

[1] $o(\cdot)$ denotes the little-o bound

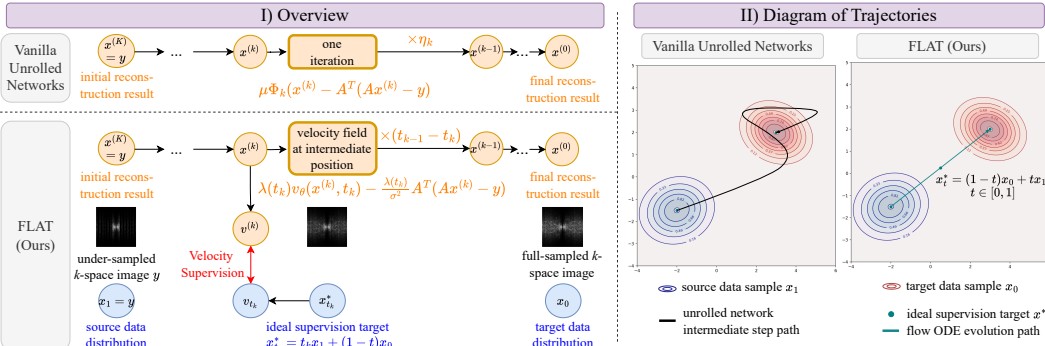

Figure 2: I) Vanilla unrolled networks vs. our Flow-Aligned Training (FLAT). Vanilla unrolled networks iteratively refine reconstructions step-by-step with supervision only at the final output. Our theory reformulates unrolling (orange) as a discretized flow ODE (blue); in FLAT, each step predicts a velocity field, with intermediate supervision that aligns predicted and ideal velocities. II) Trajectory comparison. Without intermediate supervision, vanilla unrolled networks exhibit unstable (oscillatory) trajectories that "under-run" or "overshoot" the target. FLAT supervises intermediate steps to follow stable, straight-line paths guided by flow ODE theory.

## 3.3 IMPLICATIONS OF THE THEORY

The mapping from Eq. (11) reveals that **unrolled networks correspond to implementation of descritized conditional probability flow ODE**. On one hand, unrolled networks can be interpreted as discretized flow ODE; on the other, we can implement discretized flow ODE using unrolled networks. The sequence of reconstructions $\{x^{(K-1)}, \ldots, x^{(0)}\}$ forms a discretized trajectory approximating the continuous ODE solution, providing a theoretical foundation for unrolled models. This new perspective provides a continuous-time theoretical foundation for a previously discrete and empirically-driven class of models, opening up new avenues for principled model design. From the mapping, we have the following key implications:

① **Time schedule:** Because $t$ traverses the full horizon $[1 \to 0]$, the steps $\{\delta_k\}$ must satisfy $\sum_k \delta_k = -1$. This induces a coherent, monotone cascade schedule. If $\sum_k \delta_k \neq -1$, the model reduces to a vanilla unrolled network that will either "under-run" (not reach the terminal state) or "over-run" (overshoot), manifesting as unstable or oscillatory cascades and wasted network depth.

② **Parameters grounded in ODE:** Step-size $\eta_k$ and weight $\mu$ are not free, as $\eta_k = \delta_k \lambda(t_k)/\sigma^2$ and $\mu = -\sigma^2$. Setting $\{t_k\}$ and $\lambda(t)$ determines $\{\eta_k\}$, resulting in coherent cascade sizes. This is important because if $\eta_k$ is left free (as in methods ADMM-Net, E2E-VarNet, ISTA-Net, Cascaded U-Net (Aghabiglou & Eksioglu, 2021) etc), then the effective timesteps can zig-zag or collapse, resulting in erratic intermediate images.

③ **Intermediate supervision:** Since $x^{(k)} = x_{t_k}$, supervising at intermediate timesteps with appropriate ODE-consistent targets (i.e., linearly interpolated ground truth) aligns the unrolled network with the ideal ODE trajectory, thereby improving stability and convergence.

To supervise our $K$-step network against this continuous trajectory, we select $K + 1$ discrete points for alignment. We employ a time schedule $\{t_k\}_{k=0}^K$, which is denser near $t = 0$. This sequence can be either uniform or non-uniform; empirically, a non-uniform schedule denser near $t = 0$ yields better performance. In Appendix E, we clarify that denser sampling near $t = 0$ leads to a smaller error upper bound. This provides us with a set of ideal supervisory targets $\{x_{t_k}^*\}_{k=0}^K$ sampled along the target flow. Specifically, we sample $\{t_k\}_{k=0}^K$ as:

$$t_k = 1 - (1 - k/K)^{(1+\alpha)} \tag{12}$$

where $\alpha$ is a hyperparameter controlling the density of $\{t_k\}_{k=0}^K$. The ideal targets $\{x_{t_k}^*\}_{k=0}^K$ are then computed using linear interpolation following Eq. (4).

## 3.4 FLOW-ALIGNED TRAINING STRATEGY (FLAT)

From the key implications ❶ ❷ ❸, we propose Flow-Aligned Training (FLAT), which brings flow ODEs to MRI reconstruction via unrolled networks. FLAT *(i)* enforces a time schedule by choosing a monotone sequence $\{t_k\}_{k=0}^K$ with $\delta_k = t_{k-1} - t_k$ and $\sum_k \delta_k = -1$, *(ii)* sets parameters $\eta_k$ and $\mu$ to conform to flow ODE restrictions, and *(iii)* adds intermediate supervision that aligns each cascade with the ideal ODE trajectory. Fig. 2 illustrates the contrast between vanilla unrolled networks and FLAT.

**Scheduling and hyperparameters.** We first fix the time schedule $\{t_k\}$ according to Eq. (12) to cover the full horizon $[1 \rightarrow 0]$. Given $\{t_k\}$, we compute the hyperparameters $\eta_k$ and $\mu$ directly from Eq. (11), which yields coherent cascade magnitudes and prevents zig-zag or collapsed steps.

**Velocity alignment.** The core innovation of FLAT lies in constraining intermediate steps to match the ideal ODE trajectory. We define the ideal discretized velocity at the $k$-th timestep, or the $k$-th module of the unrolled network, as the discrete temporal derivative: $v_{t_k} = \left(x_{t_k}^* - x^{(k+1)}\right) / (t_k - t_{k+1})$ where $x_{t_k}^*$ represents the ground truth linearly interpolated at time $t_k$, and $x^{(k+1)}$ is the network prediction at the $(k+1)$-th iteration. This velocity is computed using the previous iterative result as its start point. Similarly, the network's predicted velocity is: $v^{(k)} = \left(x^{(k)} - x^{(k+1)}\right) / (t_k - t_{k+1})$ where $x^{(k)}$ denotes the network's output at step $k$. The velocity alignment loss at timestep $t$ is formulated as: $\mathcal{L}_{\text{velocity}}(t_k) = |v_{t_k} - v^{(k)}|$. This velocity supervision provides a strong inductive bias that guides the network to learn physically meaningful transitions between consecutive states, leading to more stable and accurate reconstruction flows.

**Training objective.** The complete training objective combines velocity supervision $\mathcal{L}_{\text{velocity}}$ with standard reconstruction losses to ensure both dynamic correctness and reconstruction quality:

$$\mathcal{L}_{\text{FLAT}} = \sum_{k=0}^{K-1} w_k \mathcal{L}_{\text{velocity}}(t_k) + w_{\text{pixel}} \mathcal{L}_{\text{pixel}} + w_{\text{perceptual}} \mathcal{L}_{\text{perceptual}} + w_{\text{semantic}} \mathcal{L}_{\text{semantic}} \qquad (13)$$

The $L_{\text{pixel}}$ and $L_{\text{perceptual}}$ serve as anchors to maintain the visual quality of the prediction in pixel or perceptual wise. These two terms are implemented with Mean Absolute Error and $L_{\text{SSIM}}$ (Wang et al., 2004) in our experiments. These are standard losses in MRI reconstruction tasks. The $L_{\text{semantic}}$ serves as an additional term to provide semantic supervision on rich-contextual MR images, which is effective in image inverse problems (Aakerberg et al., 2022). Our $L_{\text{velocity}}$ is an additional objective to stabilize the iteration process and keep it more close to the ideal one.

## 4 EXPERIMENTS

**Datasets.** We evaluate on three public MRI datasets: Brainweb (Cocosco et al., 1997), MR-BrainS13 (Mendrik et al., 2024), and fastMRI single coil knee dataset (Zbontar et al., 2018). For all datasets, we employ 1-D equispaced fraction sampling on 2-D slices with $8\times$ acceleration and center fraction $8\%$ to simulate under-sampling. More details in Appendix B.

**Baselines and Implementation Details.** We compare against unrolled network-based methods Cascaded U-Net (Aghabiglou & Eksioglu, 2021), E2E-VarNet (Sriram et al., 2020), ReconFormer (Guo et al., 2023), MambaRecon (Korkmaz & Patel, 2025), and diffusion-based method MC-DDPM (Xie & Li, 2022). For each baseline, we use the loss terms employed in the original paper. Though FLAT is backbone-agnostic, we use E2E-VarNet as our backbone. For simplicity, we fix $\lambda(t_k) = 1$ and set $\sigma = 1$ in Eq. (10). We use $\mathcal{L}_{\text{Dice}}$ as $\mathcal{L}_{\text{semantic}}$. For hyperparameters, we set $\alpha = 4$, $w_k = 10^{-4}$, $w_{\text{pixel}} = 10$, $w_{\text{perceptual}} = 1$, and $w_{\text{semantic}} = 0.5$. More details in Appendix C.

**Evaluation Metrics.** We evaluate using Peak Signal-to-Noise Ratio (PSNR) (Hore & Ziou, 2010) and Structural Similarity Index (SSIM) (Wang et al., 2004) which are standard metrics for the MRI reconstruction task. We also perform the unpaired t-test Student (1908) (95% confidence interval) to determine the statistical significance of the improvement. The statistically significant better performances are highlighted with **bold** in all the tables. The best, while not statistically significant, performances are highlighted with *italics*.

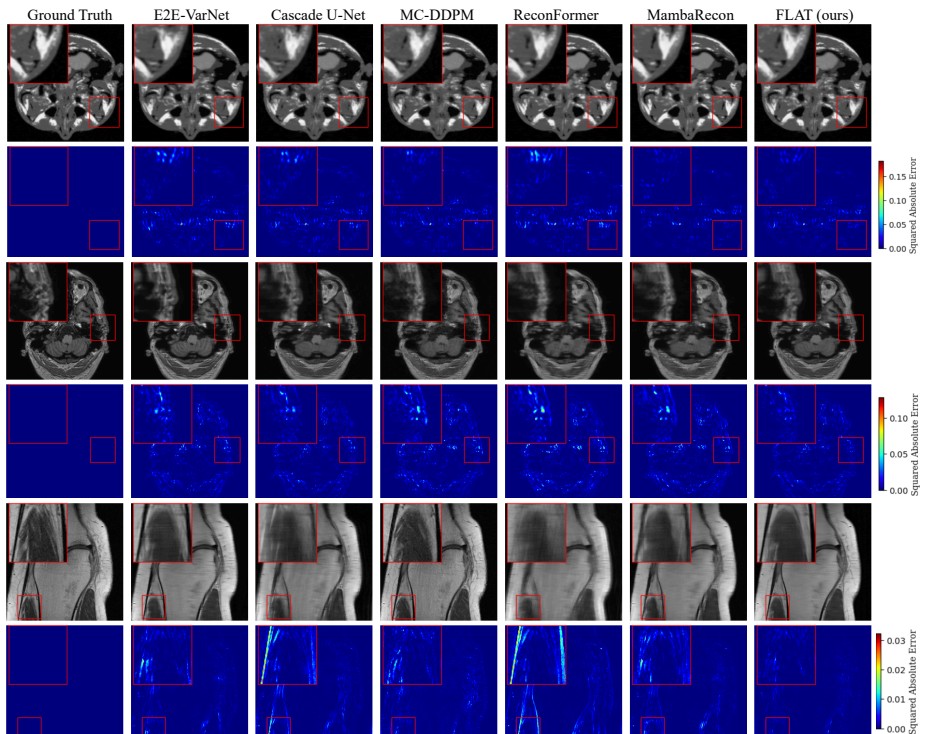

Figure 3: Qualitative results on Brainweb (rows 1-2), MRBrainS13 (rows 3-4) and fastMRI knee dataset (rows 5-6). For each dataset, the first row shows reconstructions; the second row shows the squared-error map relative to the ground truth to visualize the error magnitude.

## 4.1 RESULTS

Tab. 1 reports quantitative MRI reconstruction results across all datasets. On Brainweb and MR-BrainS13, FLAT achieves either the best statistically significant performance (**bold**) or the numerically best performance without statistical significance (*italics*). This superior performance stems from FLAT's ODE-consistent update schedule and intermediate supervision, which stabilizes the reconstruction trajectory and makes each cascade contribute meaningfully, unlike the other unrolled methods where unconstrained steps yields unstable intermediate updates. Furthermore, MC-DDPM requires 1000 diffusion steps, which is computationally expensive. In contrast, FLAT leverages the deterministic flow ODE formulation to achieve better performance with only 12 steps. On the fastMRI dataset, FLAT achieves numerically best PSNR, while the SSIM is second to E2E-Net. We further analyze this in Appendix H.

Qualitative results in Fig. 3 mirror the quantitative findings. FLAT achieves high image quality by reducing noise and artifacts, and recovering fine anatomical details compared to other methods. The squared-error maps show low-magnitude errors for FLAT vs. higher-magnitude errors for baselines.

In Tab. 2, we compare FLAT against diffusion models to understand computational efficiency (i.e., number of iterations). Both DDPM and DDIM use MC-DDPM weights. Though DDPM achieves promising performance, it requires a large number of iterative steps. DDIM improves efficiency upto 50 steps; but loses quality when compressed to a value comparable to ours (i.e., 12 steps). In contrast, FLAT is computationally efficient as it maintains image quality with just 12 steps ($3\times$ lesser than DDPM) and marginal parameter overhead, indicating that the intermediate supervision enables efficient use of depth without sacrificing fidelity.

**Takeaway.** These results indicate that FLAT successfully overcomes trajectory instability in unrolled models and avoids the iteration burden of diffusion-based sampling, delivering higher quality at low iteration counts through ODE-aligned updates and intermediate supervision.

Table 1: Comparison with existing MRI reconstruction approaches.

| Method | Brainweb | | MRBrainS13 | | fastMRI Knee | |
|---|---|---|---|---|---|---|
| | PSNR ↑ | SSIM ↑ | PSNR ↑ | SSIM ↑ | PSNR ↑ | SSIM ↑ |
| Cascaded U-Net | 31.80±2.9219 | 0.9119±0.0292 | 29.85±2.5413 | 0.9009±0.0484 | 31.01±3.3856 | 0.6749±0.1390 |
| E2E-VarNet | 31.88±3.0380 | 0.9194±0.0266 | 30.40±2.7764 | 0.9212±0.0451 | 31.39±3.4799 | **0.7025±0.1346** |
| MC-DDPM | 32.52±3.9800 | 0.9194±0.0348 | 28.15±2.3492 | 0.8588±0.0554 | 29.39±3.5800 | 0.5785±0.1713 |
| ReconFormer | 30.75±3.3964 | 0.8984±0.0587 | 30.22±2.5339 | 0.8683±0.0503 | 30.55±3.4174 | 0.6691±0.1398 |
| MambaRecon | 33.25±3.2946 | 0.9219±0.0764 | 28.62±2.3664 | 0.8816±0.0541 | 27.22±3.0227 | 0.5474±0.1647 |
| FLAT (ours) | **33.52±3.3125** | **0.9395±0.0267** | **33.41±3.0455** | *0.9259±0.0427* | *31.44±3.6524* | 0.6789±0.1493 |

Table 2: Comparison against diffusion-based models with different number of steps.

| Method | # of iterative steps | # of model parameters | PSNR ↑ | SSIM ↑ |
|---|---|---|---|---|
| DDPM | 1000 | 81M | 32.52±3.9800 | 0.9194±0.0348 |
| DDIM | 50 | 81M | 31.00±2.2192 | 0.8313±0.0326 |
| DDIM | 12 | 81M | 21.66±0.6338 | 0.3819±0.0905 |
| FLAT (ours) | 12 | 93M | **33.52±3.3125** | **0.9395±0.0267** |

## 4.2 ABLATION STUDIES

To demonstrate the efficacy of FLAT, we conduct comprehensive ablation studies on the Brainweb dataset. We analyze individual contributions and trajectory stability. Due to space constraints, we discuss hyperparameter sensitivity, loss term impact, $\sigma$ value impact and MRI acceleration level impact in Appendix D.

**Impact of Components.** In Tab. 3, we evaluate two key components of FLAT: ❶-❷ explicitly setting the ODE-derived hyperparameters $\{\delta_k\}_{k=1}^{K}$, $\{\eta_k\}$ and $\mu$, versus ❸ using intermediate (velocity) supervision along the ideal ODE trajectory. In the first row, $\{t_k\}_{k=0}^{K}$ is left unspecified and is learned implicitly, following Sriram et al. (2020). In the second row, $\{t_k\}_{k=0}^{K}$ is explicitly

Table 3: Ablation on ❶❷❸.

| ❶-❷ | ❸ | PSNR ↑ | SSIM ↑ |
|---|---|---|---|
| ✗ | ✗ | 32.86±3.4841 | 0.9333±0.0303 |
| ✓ | ✗ | 32.83±3.2937 | 0.9325±0.0275 |
| ✗ | ✓ | 33.14±3.4614 | 0.9356±0.0287 |
| ✓ | ✓ | **33.52±3.3125** | **0.9395±0.0267** |

set per Eq. (12), with $\sum_k \delta_k = -1$. In the third row, supervising intermediates without an explicit $\{t_k\}$ effectively collapses to supervising toward the final target $x_0$ at each step. From Tab. 3, it is clear that the combination (FLAT, row 4) yields the largest gain by fully leveraging the flow-ODE perspective.

**Image Quality During Evolution.** We analyze reconstruction quality across iterations in Fig. 4. The evolution is from step 12 (initialization) to step 0 (final output), with the first output at step 11. The unrolled baseline attains good final PSNR but exhibits unstable (fluctuating) intermediate outputs, indicating that only the final step contributes meaningfully while earlier cascades remain underutilized. Diffusion mod-

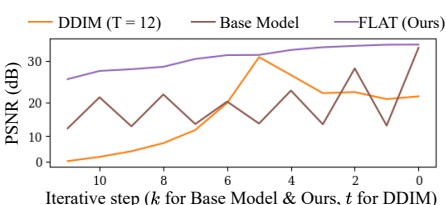

Figure 4: PSNR curves for 12-step iterations.

els reach strong quality with many steps but struggle with over-denoising artifacts when compressed to 12 steps. In contrast, FLAT maintains high PSNR throughout the sequence and shows smooth, monotonic improvements, indicating both better intermediate quality and more stable evolution.

**Image Quality Across Timesteps.** We evaluate image quality at each timestep. We compare against diffusion models, because vanilla unrolled methods do not have an explicit schedule for $\{t_k\}_{k=0}^{K}$, and so their time indices are undefined. In Fig. 5, DDPM exhibits a smooth PSNR–$t$ curve but requires large number of timesteps. DDIM requires fewer timesteps, but has a lower PSNR overall, especially when the number of timesteps is very small. In contrast,

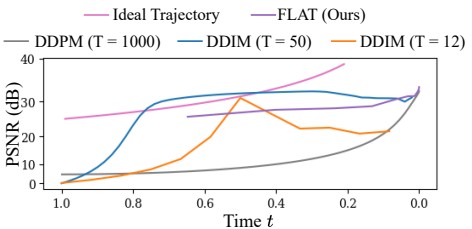

Figure 5: PSNR curves for timestep $t = 1 \to 0$.

FLAT stays close to the ideal trajectory while using only 12 discrete timesteps. The evolution progresses from $t = 1$ to $t = 0$ and demonstrates FLAT's efficiency in achieving stable convergence with $3\times$ lesser steps compared to DDPM.

## 5 CONCLUSION

In this work, we introduce flow ODEs to MRI reconstruction by theoretically proving that unrolled networks are discrete implementations of conditional probability flow ODEs. This connection reveals that effective unrolled training (i) requires an explicit time schedule, (ii) ODE-grounded hyperparameter constraints, and (iii) intermediate supervision. Building on this, we introduced Flow-Aligned Training (FLAT), which explicitly sets hyperparameters and aligns cascade updates with the ideal ODE trajectory to stabilize training and improve convergence. Across three MRI datasets, FLAT delivers high-quality reconstructions with up to $3\times$ fewer iterations compared to diffusion models, and markedly more stable intermediate behavior than conventional unrolled baselines.

## 6 REPRODUCIBILITY STATEMENT

For reproducibility, we provide dataset and implementation details in Sec. 4, with detailed descriptions in Appendix B and Appendix C.

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

Appendix A clarifies the use of Large Language Models in manuscript preparation.

Appendix B provides detailed descriptions of the datasets used in our experiments.

Appendix C provides the detailed implementation of baselines and FLAT.

Appendix D presents ablation study on hyperparameters $\alpha$ and $K$.

Appendix E provides a proof of why denser $\{t_k\}_{k=1}^{K}$ near $k = 0$ reach smaller error upper bounds of discretized flow ODE.

Appendix F provides some discussion of the ideas in this work.

Appendix G discusses the limitations.

Appendix H discusses the experimental results on fastMRI dataset.

Appendix I discusses the wall clock inference time comparison among different methods.

## A  USE OF LARGE LANGUAGE MODELS

We used large language models (LLMs) solely as an assistive tool for grammar refinement and improving the clarity of writing. We did not use it to find related work or for research ideation. Hence, all authors take full responsibility for the content of this paper.

## B  DATASET DESCRIPTION

**Brainweb (Cocosco et al., 1997).** This is a publicly available MR brain image simulation tool, which provides clear-structured MR images. We synthesized 20 T1-weighted brain MR image volumes with a voxel resolution of 1 mm. Each volume consists of 362 slices. All slices are cropped to $256 \times 256$. A 10/5/5 train/val/test data split was used for this dataset. The number of 2-D slices for training, validation and testing are 3620, 1810 and 1810 respectively.

**MRBrainS13 (Mendrik et al., 2024).** This dataset consists of 20 MR imaging cases. We only use T1-weighted MR image volumes in our experiments. Each volume has a voxel size $0.96 \times 0.96 \times 3\text{mm}^3$, and contains 48 slices. All slices are cropped to $224 \times 224$. We split this dataset into train, val and test sets, respectively containing 5, 7 and 8 volumes. The number of 2-D slices for training, validation and testing are 240, 336 and 384 respectively.

**fastMRI Knee (Zbontar et al., 2018).** We use the single coil data from this dataset. To obtain ground truth data in $k$-space, we only use training set. Inside this set, there are 973 volumes in total. The number of slices in each volume ranges from 28 to 50. The in-plane resolution is 0.5mm $\times$ 0.5mm, and the slice thickness is 3mm. 486, 195 and 292 volumes are used for training, validation and testing, respectively. All slices are cropped to size $320 \times 320$. The number of 2-D slices for training, validation and testing are 17287, 6945 and 10510 respectively.

## C  IMPLEMENTATION DETAILS

**Implementation of the Base Model.** Though our FLAT is backbone-agnostic, we use E2E-VarNet (Sriram et al., 2020) as our base model. In our base model, the implementation of $v_\theta(x_{t_k}, t_k)$ is as follows:

$$v_\theta(x_{t_k}, t_k) = A^T \left( A\mathcal{F} \circ \mathcal{E} \circ \text{CNN} \left( \mathcal{R} \circ \mathcal{F}^{-1}(x^{(k)}) \right) - y \right) \tag{14}$$

where $A$ is the sampling matrix in $k$-space, $\mathcal{F}$ is the Fourier Transform, $\mathcal{F}^{-1}$ is the Inverse Fourier Transform, $\mathcal{R}$ is the Root Sum-of-Squares which is the reduction from multi coil to single coil, $\mathcal{E}$ is the expansion from single coil to multi coil. Both $\mathcal{R}$ and $\mathcal{E}$ are computed according to a sensitivity map, so we have a distinct network to estimate it, which follows E2E-VarNet (Sriram et al., 2020).

**Baselines.** We compare against baselines Cascaded U-Net (Aghabiglou & Eksioglu, 2021), E2E-VarNet (Sriram et al., 2020), MC-DDPM (Xie & Li, 2022), ReconFormer (Guo et al., 2023) and

Figure 6: Impact of hyperparameters $\alpha$ and $K$.

Table 4: Impact of Loss Terms.

| $L_{velocity}$ | $L_{perceptual}$ | $L_{pixel}$ | $L_{semantic}$ | PSNR | SSIM |
|---|---|---|---|---|---|
| | ✓ | | | 33.06±3.3442 | 0.9361±0.0282 |
| ✓ | ✓ | | | 33.62±3.3752 | 0.9412±0.0269 |
| | ✓ | ✓ | | 33.15±3.2981 | 0.9350±0.0277 |
| ✓ | ✓ | ✓ | | 33.44±3.3476 | 0.9378±0.0275 |
| | ✓ | ✓ | ✓ | 32.83±3.2937 | 0.9325±0.0275 |
| ✓ | ✓ | ✓ | ✓ | 33.52±3.3125 | 0.9395±0.0267 |

MambaRecon (Korkmaz & Patel, 2025). For each baseline, we use the loss terms employed in the original paper.

**Hyperparameters.** For simplicity, we fix $\lambda(t_k) = 1$ and set $\sigma = 1$ in Eq. (10). We use Mean Absolute Error (MAE) as $\mathcal{L}_{\text{pixel}}$, Structural Similarity (SSIM) (Wang et al., 2004) loss as $\mathcal{L}_{\text{perceptual}}$, and $\mathcal{L}_{\text{Dice}}$ as $\mathcal{L}_{\text{semantic}}$. We compute the Dice loss on segmentation results, which is estimated using unsupervised Gaussian Mixture Model (GMM). We set $\alpha = 4$. To balance all loss terms to the same scale, we set $w_k = 10^{-4}$, $w_{\text{pixel}} = 10$, $w_{\text{perceptual}} = 1$, and $w_{\text{semantic}} = 0.5$. We trained our network from scratch with AdamW optimizer, using learning rate $10^{-3}$ and batch size 1, which are same with our base network E2E-VarNet. We trained for 200 epochs on a single NVIDIA A5000 GPU. We will publicly release the code upon acceptance of the paper.

**Unsupervised Segmentation.** We compute the segmentation results using unsupervised Gaussian Mixture Model (GMM). Specifically, for each complex-value volume in image space, we first compute absolute value for all voxels. Then, 100,000 voxels are randomly sampled to estimate GMM model. We set the number of clusters for GMM to 4. For each volume there will be a distinct GMM model. After estimation, we use this model to compute the segmentation likelihood map on both fully-sampled and reconstructed image, and compute soft Dice loss.

# D    ADDITIONAL ABLATION STUDY

**Impact of $\alpha$ and $K$.** Fig. 6 examines the impact of timestep-density factor $\alpha$ and number of steps $K$. As expected, increasing $K$ provides more iterations for refinement, leading to improved reconstruction quality. Similarly, larger $\alpha$ values yield better image quality by concentrating more timesteps near $t = 0$, where fine-scale refinement occurs. This distribution is crucial because our analysis reveals that most reconstruction steps focus on refinement rather than denoising—only a limited number of initial steps are needed to produce visually acceptable images. Therefore, a larger $\alpha$ allocates more computational resources to the refinement phase, while smaller $\alpha$ values under-utilize refinement steps, resulting in degraded image quality. We use these two experiments to select the $\alpha$ and $K$ value, and use the same combination of $\alpha$ and $K$ across different datasets.

**Impact of Loss Terms.** We compared the impact of all four loss terms in Tab. 4. As expected, $L_{\text{perceptual}}$, $L_{\text{pixel}}$ and $L_{\text{semantic}}$ contribute to the visual performance, and $L_{\text{velocity}}$ contributes to additional PSNR and SSIM improvement.

**Impact of $\sigma$** We tested the impact of $\sigma$ in Tab. 5. We claim that the $\sigma$ value is a normalized scale factor, and the value of $\sigma$ slightly impacts the reconstruction performance. For simplicity, we select $\sigma = 1$.

Table 5: Impact of Parameter $\sigma$.

| $\sigma$ | PSNR | SSIM |
|---|---|---|
| 0.25 | 33.44±3.3331 | 0.9375±0.0281 |
| 0.5 | 33.29±3.4089 | 0.9359±0.0285 |
| 1 | 33.52±3.3125 | 0.9395±0.0267 |
| 2 | 33.44±3.3726 | 0.9368±0.0281 |

Table 6: Impact of MRI acceleration level.

| Acc | w/ ours | PSNR | SSIM |
|---|---|---|---|
| 4 | | 41.34±3.6783 | 0.9830±0.0086 |
| 4 | ✓ | 42.08±3.5000 | 0.9859±0.0076 |
| 8 | | 32.86±3.4841 | 0.9333±0.0303 |
| 8 | ✓ | 33.52±3.3125 | 0.9395±0.0267 |
| 12 | | 30.45±3.6086 | 0.8995±0.0408 |
| 12 | ✓ | 30.66±3.5985 | 0.9050±0.0393 |

**Impact of MRI Acceleration Level.** Tab. 6 examines the impact of our approach on a various of MRI acceleration levels. When the acceleration level is getting higher, i.e. the image quality is lower, the increasement of reconstruction performance gets slighter, as the network capacity is limited.

# E   CLARIFICATION OF DENSER $t_k$ NEAR $t = 0$ REACHES SMALLER ERROR UPPER BOUND

The Taylor expansion of Eq. (5) is

$$x_{t_{k-1}} = x_{t_k} + \delta_k \frac{dx}{dt}|_{t=t_k} + \frac{\delta_k^2}{2}\frac{d^2x}{dt^2}|_{t=\xi_k}, \xi_k \in [t_{k-1}, t_k] \tag{15}$$

our discretized formulation Eq. (10) is

$$x_{t_{k-1}} = x_{t_k} + \delta_k \frac{dx}{dt}|_{t=t_k} \tag{16}$$

implemented using neural network, our formulation becomes:

$$x^{(k-1)} = x^{(k)} + \delta_k f(t_k, x^{(k)}) \tag{17}$$

where $f(\cdot, \cdot)$ is our implementation of $\frac{dx}{dt}$. Define the error as:

$$e_k = \|x_{t_k} - x^{(k)}\| \tag{18}$$

the actual $\frac{dx}{dy}$ as:

$$g(x, t) := \frac{dx}{dt} \tag{19}$$

and the error of $f(\cdot, \cdot)$ as:

$$\eta_k = \|g(t_k, x_{t_k}) - f(t_k, x_{t_k})\| \tag{20}$$

then we have

$$
\begin{aligned}
e_{k-1} &= \|x_{t_{k-1}} - x^{(k-1)}\| \\
&= \|\left[x_{t_k} + \delta_k g(x_{t_k}, t_k) + \frac{\delta_k^2}{2}\frac{d^2x}{dt^2}|_{t=\xi_k}\right] - \left[x_k + \delta_k f(t_k, x^{(k)})\right]\| \\
&= \|[x_{t_k} - x^{(k)}] + \delta_k[g(x_{t_k}, t_k) - f(t_k, x^{(k)})] + \frac{\delta_k^2}{2}\frac{d^2x}{dt^2}|_{t=\xi_k}\|
\end{aligned}
\tag{21}
$$

According to Lipschitz continuity, for $f(t, x)$ we have a non-negative $L(t_k)$ such that:

$$\|f(t_k, x_1) - f(t_k, x_2)\| \le L(t_k)\|x_1 - x_2\| \tag{22}$$

And according to Triangle inequality we have:

$$\|g(t_k, x_{t_k}) - f(t_k, x^{(k)})\| \leq \|g(t_k, x_{t_k}) - f(t_k, x_{t_k})\| + \|f(t_k, x_{t_k}) - f(t_k, x^{(k)})\|$$
$$\leq \eta_k + L(t_k)e_k \quad (Eq.~(22)) \tag{23}$$

Combine Eqs. (21) and (23) together yielding:

$$e_{k-1} \leq e_k + |\delta_k|(L(t_k)e_k + \eta_k) + \frac{\delta_k^2}{2}\|\frac{d^2x}{dt^2}|_{t=\xi_k}\| \tag{24}$$

For simplification, we define the upper bound of $\frac{d^2x}{dt^2}$:

$$M(t_k) = \sup_{t \in [t_{k-1}, t_k]} \|\frac{d^2x}{dt^2}\| \tag{25}$$

Then we have:

$$e_{k-1} \leq e_k(1 + L(t_k)|\delta_k|) + |\eta_k|\delta_k + \frac{\delta_k^2}{2}M(t_k) \tag{26}$$

Consequently, the upper bound of error $e_0$ at final iterative step $k = 0$ is the accumulation of three terms: the error term $(L(t_j)\delta_j)$, the error term of $f(\cdot, \cdot)$: $|\delta_k|\eta_k$ and the maximum local error term $\frac{\delta_k^2}{2}M(t_k)$.

Note that, the early terms ($k \to 0$) impacts the entire accumulation. Therefore, it is reasonable to reduce the error by using smaller $|\delta_k|$ when $k \to 0$. That is, when $t$ is close to 0, we need to use a smaller $\delta_k$, and use a denser of $t$.

## F    DISCUSSION

**Necessity of Explicit Timestep Control.** Explicit timestep control in Eq. (10), or explicit step size in Eq. (3), stabilizes the evolution. As shown in Fig. 4 and Fig. 5, without explicitly controlling the timesteps, the intermediate steps of unrolled networks contribute very little to the final reconstruction; it is only the last step that contributes the most. This suggests unstable iterations in Eq. (10) or Eq. (3). We believe the lack of explicitly set timesteps causes this issue. Theoretically, the sum of all $|t_{k-1} - t_k|$ in Eq. (10) should equal to 1, therefore an explicit timestep constraint is necessary for stable evolution. To prove this, we add this constraint to unrolled networks and show the results in Fig. 4. We observe that these methods appear to have more stable evolutions.

**Explicit Intermediate Step Control in Iterative Networks.** We expand the above to iterative networks such as diffusion models. Referring Fig. 4 and Fig. 5, we demonstrate that, for iteration-based networks, it is critical to constrain the intermediate steps to follow an ideal trajectory. Such constraint pushes the evolution path to the ideal one, thereby improving the final reconstruction performance. We also demonstrate that the reason the performance of DDIM deteriorates when number of iterations is low (e.g. 12) is because the evolving trajectory strays far away from the theoretically ideal path.

**Deterministic Image Generation.** Our ODE-based approach does not introduce any randomness in the flow evolution, which differs from SDE-based image generation methods like diffusion. We believe that this is determined by type of images and imaging techniques, and we hardly need randomness for medical imaging techniques such as MRI. Suppose we scan the MRI for a knee. A knee is just a knee, and there is no randomness inside a knee. There will be some randomness introduced during imaging, but the ideal imaging technique should not contain any randomness for this knee. Therefore, we argue that a deterministic image generation method is more suitable for this kind of medical imaging task.

## G    LIMITATIONS

We use unidirectional velocity supervision (Eq. (10)) to estimate $x_{t_{k-1}}$ from $x_{t_k}$. Further study can explore bidireational flow evolving paths to better fit the ideal trajectory. Following Sriram

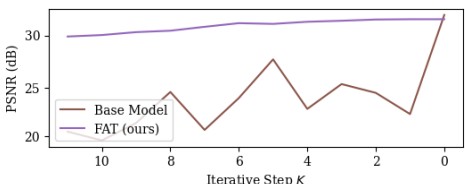

Figure 7: Change of PSNR value among different step $K$ on fastMRI knee single coil dataset.

Table 7: Experiments on fastMRI knee multi-coil dataset.

| w/ ours | PSNR | SSIM |
|---|---|---|
| | 29.10±4.1675 | 0.8002±0.1144 |
| ✓ | 29.02±4.4695 | 0.7913±0.1218 |

et al. (2020), we use a simple Norm U-Net as our backbone for estimating $v_\theta$ in Eq. (10). A more robust predictor such as ViT (Han et al., 2022; Liu et al., 2021) and Mamba (Gu & Dao, 2023) may provide a more refined image. Further, we design our ideal evolution path of flow ODE in $k$-space. Future study can explore a more robust solution to evolve in latent space to maintain high-level image features. Our analysis only adjust to one-order gradient based unrolled networks, and cannot explicitly cover variable splitting based methods such as ADMM-Net. Expanding our analysis to ADMM will be an interesting and non-trivial work. Finally, while we demonstrate that our method can be extended to other data generation tasks as long as the data degradation kernel $A$ (from Eq. (1) is a known value, in this work, we focus our experiments on the MRI reconstruction task. In a future study, we would like to conduct experiments on other tasks.

## H    RESULTS ON FASTMRI KNEE

Fig. 7 illustrates the PSNR value in iteration on base network and ours on fastMRI knee single coil dataset. The evolution is from step 12 (initialization) and step 0 (final output). Though our FLAT does not achieves the best SSIM, the numerical value is ordered in the second places. Compared to the E2E-VarNet (or the base model) which achieves the best SSIM, Fig. 7 shows that only the final step contributes meaningfully, while early steps are underutilized. On the contrary, our FLAT shows a increasing reconstruction performance improvements, indicating better intermediate stability. We also conduct experiments on fastMRI knee multi coil dataset (Zbontar et al., 2018) in Tab. 7, which shows similar results with single coil data.

## I    WALL CLOCK INFERENCE TIME

We test wall clock inference time among different approaches on Brainweb dataset to further explain the speed. The input image size is $256 \times 256 \times 2$, where 2 indicates that the image values are complex. For FLOPS of DDPM and DDIM, we tested the single step FLOPS, and multiplied by the number of steps to compute the overall FLOPS. Our approach has the same FLOPS and wall clock inference time with the base network, while existing generative models for MRI reconstruction (DDPM or DDIM based) requires a much higher inference time.

Table 8: Wall clock time for inference.

| Model | FLOPS | Inference Time |
|---|---|---|
| Cascaded U-Net | 68G | 22ms |
| E2E-VarNet | 165G | 80ms |
| MC-DDPM (DDPM) | $273G \times 1000$ | 48319ms |
| MC-DDPM (DDIM) | $273G \times 50$ | 5724ms |
| ReconFormer | 173G | 222ms |
| Ours | 165G | 88ms |

