# OpenReview forum: "Unrolled Networks are Conditional Probability Flows in MRI Reconstruction"
_ICLR.cc/2026/Conference — Submitted to ICLR 2026_

### Official Review · Reviewer_hR2V · 2025-10-30

**Soundness:** 3
**Presentation:** 4
**Contribution:** 4
**Rating:** 6
**Confidence:** 5

**Summary:**

The paper addresses accelerated MRI reconstruction from under-sampled k-space by establishing a theoretical equivalence between unrolled networks and conditional probability flow ODEs. It demonstrates that each cascade in an unrolled model corresponds to a forward-Euler step of a conditional probability flow, thereby imposing explicit constraints on the timestep schedule and model hyperparameters. This formulation provides a clear and useful theoretical perspective—viewing unrolled MRI reconstruction as a discretized conditional probability flow—which naturally leads to an ODE-grounded training scheme that is both simple to implement and empirically effective. Building upon this insight, the authors introduce Flow-Aligned Training (FLAT), which (i) enforces ODE-consistent timestep scheduling, (ii) fixes step sizes and weights through the ODE mapping, and (iii) introduces intermediate “velocity alignment” supervision via a composite loss. Experiments show that FLAT achieves higher or comparable PSNR and SSIM using only 12 steps, outperforming diffusion-based baselines that typically require 50–1000 iterations.

**Strengths:**

- **Unified view of unrolled networks and probabilistic flow models.**
  The paper presents a significant conceptual advance by unifying *unrolled reconstruction networks* and *conditional probabilistic flow ODEs*. It formally shows that the cascaded updates in unrolled networks can be interpreted as **forward Euler discretizations** of conditional probability flows, thus providing a clear theoretical framework for understanding unrolled MRI reconstruction.
  This perspective addresses several long-standing issues in unrolled models, including:
  - Unstable or redundant reconstruction trajectories with unclear intermediate meanings.
  - Poor interpretability due to backpropagation-based hyperparameter tuning.

By redesigning the unrolled structure from a **probabilistic flow** perspective, the paper achieves better interpretability and stability while maintaining strong reconstruction performance. This work offers conceptual value not only to the unrolled MRI community but also to the broader field of deep iterative reconstruction, potentially inspiring more interpretable and theoretically grounded iterative architectures.

**Weaknesses:**

- **Inadequate diffusion-based baseline comparison.**
  The comparison against diffusion-based methods is insufficient. The paper claims that FLAT outperforms traditional unrolled networks and is faster and better than diffusion-based approaches. However, the chosen baseline—**MC-DDPM**—is not representative or competitive. I strongly recommend including at least one **SOTA diffusion-based method**, such as **DDS [1]**, to substantiate the claimed advantages of FLAT over diffusion models.


> [1] Chung, Hyungjin, Suhyeon Lee, and Jong Chul Ye. **"Decomposed Diffusion Sampler for Accelerating Large-Scale Inverse Problems."** *ICLR*, 2024.

---
While the proposed idea is novel and promising, the experimental section requires further strengthening. More robust comparisons and additional analyses (especially addressing the questions below) are necessary to convincingly validate the method. With these improvements, I would be very willing to reconsider my evaluation positively.

**Questions:**

- **Line 226 clarification:** The statement “conditional flow ODE evolving from the under-sampled initialization $x_1 = y$ towards the fully-sampled $x_0$” is not precise. Since $y$ denotes under-sampled k-space measurements, the initialization should more appropriately be $x_1 = A^\top y$.

- **Performance on fastMRI Knee dataset:**
  Table 1 shows that FLAT does not significantly outperform traditional unrolled networks on the fastMRI Knee dataset and even underperforms in some metrics. Could the authors clarify the reason for this performance gap?

- **Complex loss function design:**
  The proposed training involves a highly complex loss composition. Could the authors analyze the contribution of each component and whether all terms are necessary?

- **Ablation interpretation (Table 3):**
  From Table 3, removing the paper’s three main contributions theoretically reduces FLAT to an E2E-VarNet with the proposed complex loss. The results show that this “degraded” model still performs notably better than a standard E2E-VarNet, suggesting that the improvement may largely stem from the loss function itself. If so, would applying the same loss function to a vanilla E2E-VarNet yield even higher performance—perhaps surpassing FLAT—on the fastMRI Knee dataset?

---

> ### Author Response · Authors · 2025-11-27
> **Response to Reviewer hR2V**
>
> We thank the reviewer for the encouraging and insightful comments. Please find our responses to specific queries below.
>
> **Q1: Comparing with DDS**
>
> **A1**: We have conducted experiments of DDS on three datasets, and reported the results as following. The performance is worse than our FLAT shown in Tab. 1 (Sec 4.1, Page 9).
>
> | Dataset | PSNR         | SSIM          |
> |---------|--------------|---------------|
> | Brainweb | 27.12±3.5318 | 0.7968±0.0624 |
> | MRBrainS13 | 27.89±2.1891 | 0.8055±0.0635 |
> | fastMRI | 30.06±3.2911 | 0.6115±0.1383 |
>
> **Q2: Line 226 clarification.**
>
> **A2**: This notation has caused some misunderstanding. Clearly, we use the zero-filled pseudo reconstruction for initialization, i.e. $x^{(k)}=A^Ty$.
>
> **Q3: Performance on fastMRI Knee dataset.**
>
> **A3**: On the fastMRI Knee dataset, our results are slightly lower than the best baseline E2E-VarNet in SSIM. We do not claim SOTA metrics over unrolled networks in the paper.The benefit of our FLAT mainly lies in stable trajectories. In Fig. 7 (Appendix H, Page 17), we visualize the PSNR values of intermediate steps in fastMRI dataset. The PSNR value of intermediate steps of E2E-VarNet is unstable, implying a zig-zag evolve trajectory in unrolled networks. On the contrary, our FLAT achieves a continuously increasing PSNR value, and has a very erratic trajectory.
>
> **Q4: Complex loss function design.**
>
> **A4**: The three primary loss terms, including $L_{pixel}$, $L_{perceptual}$ and $L_{semantic}$, are inherited from existing MRI reconstruction literatures. Only the $L_{velocity}$ term is our contribution. We have conducted ablation analysis on these loss terms to illustrate the effectiveness of $L_{velocity}$ in the Global Response. Please refer Global Response for details of the experiments and analysis.
>
> **Q5: Ablation interpretation (Tab. 3).**
>
> **A5**: We admit that introducing different loss terms impacts the reconstruction performance. Utilizing the primary loss term without $L_{velocity}$, while maintaining a promising reconstruction result, gets a worse metric compared with our approach. We have conducted ablation experiments to analyze how the primary loss term impacts the reconstruction results, and how $L_{velocity}$ improves the performance with different primary loss terms. Please refer to the Global Response for details of experiments and analysis.

---

### Official Review · Reviewer_nr56 · 2025-10-31

**Soundness:** 3
**Presentation:** 3
**Contribution:** 3
**Rating:** 8
**Confidence:** 3

**Summary:**

This paper introduces a new method (FLAT) for viewing unrolled reconstruction approaches as flow models. This insight allowed led to three critical changes in training unrolled networks for reconstruction (1) unrolls viewed as cascaded time steps must satisfy constrains as a trajectory (2) normally free hyper parameters in unrolled methods are fixed to satisfy ODE (3) at intermediate time steps (unrolls) the images are aligned to the desired trajectory not just the final image. The authors show through several experiments that enforcing these constraints on their unrolled network led to SOTA performance compared to other existing methods in MRI reconstruction.

**Strengths:**

Overall, this paper provides a novel and useful insight into existing unrolled techniques which I think would be of interest to the broader MRI recon community since unrolled techniques are very popular. The connection between flows and unrolls is interesting and clearly improves performance which is great. The authors did a good job comparing to other SOTA methods.

**Weaknesses:**

I do believe that the paper would benefit from testing their method on various acceleration levels of MRI data. They show results for R=8 but I would also like to see what their performance gains are at higher (and) lower acceleration levels like R=4 and 12. Additionally I would like to know what the wall clock time is for running inference of their method vs. the other methods presented.  They present the number of iterations compared to other techniques, but it would be nice to see the actual timing.

**Questions:**

1.	I am not sure about the statement in lines 238-239 where it is stated that $p(y|x_t)=N(y|Ax_t,\sigma^2)$. Isn’t this only true for t=0?
2.	Is the initial reconstruction $x^k = y$ the pseudo-inverse reconstruction?

---

> ### Author Response · Authors · 2025-11-27
> **Response to Reviewer nr56**
>
> We thank the reviewer for the encouraging and insightful comments. Please find our responses to specific queries below.
>
> **Q1: Testing on various acceleration levels of MRI data.**
>
> **A1**: We have conducted experiments on various MRI acceleration levels including 4, 8, and 12. This table has been added to the manuscript as Tab. 6 in Appendix D, Page 16. The result shows that our FLAT leads to performance improvement on all acceleration levels. When the acceleration level is too high (e.g. 12), the reconstruction quality is low, limiting the performance improvement.
>
> | Acc | w/ ours | PSNR         | SSIM          |
> |-----|---------|--------------|---------------|
> | 4   |         | 41.34±3.6783 | 0.9830±0.0086 |
> | 4   | ✓       | 42.08±3.5000 | 0.9859±0.0076 |
> | 8   |         | 32.86±3.4841 | 0.9333±0.0303 |
> | 8   | ✓       | 33.52±3.3125 | 0.9395±0.0267 |
> | 12  |         | 30.45±3.6086 | 0.8995±0.0408 |
> | 12  | ✓       | 30.66±3.5985 | 0.9050±0.0393 |
>
>
>
> **Q2: Time for running inference**
>
> **A2**: Ideally the inference time is the same as the base network (E2E-VarNet) because we do not modify the model architecture. We have compared the inference time of different methods, and report the results in the following table. The comparison was conducted on the Brainweb dataset, with input image size $256\times 256$. For DDPM and DDIM  results, we tested the single step FLOPS, and multiplied by the number of steps to compute the overall FLOPS. This table has been added in the manuscript as Tab. 8, Appendix I in Page 19. The diffusion models (DDPM and DDIM) take the longest inference time and largest FLOPS value. MambaRecon is the fastest method due to its efficient network architecture. Our approach has the same FLOPS and similar single slice inference time with E2E-VarNet. This result is under expected, because ours and E2E-VarNet have identical network architecture.
>
> | Model          | FLOPS              | Inference Time |
> |----------------|--------------------|----------------|
> | Cascaded U-Net | 68G                | 22ms           |
> | E2E-VarNet     | 165G               | 80ms           |
> | MC-DDPM (DDPM) | 273G $\times$ 1000 | 48319ms        |
> | MC-DDPM (DDIM) | 273G $\times$ 50   | 5724ms         |
> | ReconFormer    | 173G               | 222ms          |
> | MambaRecon    | 3G               | 14ms         |
> | Ours           | 165G               | 88ms           |
>
>
> **Q3: I am not sure about the statement in lines 238-239 where it is stated that $p(y|x_t)=N(y|Ax_t, \sigma^2)$. Isn’t this only true for t=0?**
>
> **A3**: Thanks for raising your concern. This is due to a too strong assumption. We now fix this problem, and the impact to our theory is very limited. We simply switch from Bayes’ posterior to an energy function, and there is no impact on the flow ODE formulation. Please refer to the response to **Q1.1** of reviewer jvCY for details.
>
> **Q4: Is the initial reconstruction $x^{k}=y$  the pseudo-inverse reconstruction?**
>
> **A4**: We realize that this notation has caused some misunderstanding. We use the zero-filled pseudo reconstruction for initialization, i.e. $x^{(k)}=A^Ty$.

---

### Official Review · Reviewer_jvCY · 2025-10-31

**Soundness:** 1
**Presentation:** 2
**Contribution:** 2
**Rating:** 2
**Confidence:** 5

**Summary:**

The paper aims to develop a flow ODE characterization for unrolled networks. The idea is interesting, but unfortunately there are fundamental flaws.

**Strengths:**

- I think the overall idea of using a flow ODE characterization to describe unrolled networks is great. Hence I really wanted to like this paper. Unfortunately both the theory and execution has substantial flaws.

**Weaknesses:**

1) The proof of the main result is fundamentally flawed:
- The argument hinges on writing out p(y|x_t). Unfortunately Eq. 1 does not apply to intermediate points on the trajectory, which is well-known in the literature. p(y|x_t) would need to be calculated as (in the authors' notation): \int p(y|x_0) p(x_0| x_t) dx_0, since we only know the relationship between y and x_0 (i.e. Eq. 1). This breaks down the whole proof. There are many works on approximating this integral in the diffusion inverse problems literature.
- The authors can see this fails by considering their own definition of x1 = y (incidentally I'm surprised they are trying to come up with a velocity field from k-space to image domain)
- The second part of the proof that is questionable is the statement "this velocity aligns with the gradient of the conditional log-density" Why is this true? This is not shown.
- Also fundamentally one would expect (5) to have dependence on y in this setup, instead (5) is describing an unconditional flow on the image set, with no knowledge of measurements.

2) The authors seem to be unaware of MRI reconstruction literature. Much of the motivational claims are untrue or incomplete:
- "hyperparameters such as step size and weighting coefficients are typically set through heuristics or empirical tuning"
Almost all algorithm unrolling frameworks learn step sizes and weighting coefficients jointly with the proximal operator/regularization neural network. This statement is therefore incorrect. Furthermore, these are considered parameters of the unrolled network, not hyperparameters. This joint learning of regularization and data fidelity parameters is the main advantage of algorithm unrolling over plug-and-play type methods.
- "they are typically trained with supervision only at the final cascade"
This is partially true. First off, it seems the authors are not distinguishing between unrolled networks with shared parameters (i.e. each cascade uses the same CNN and step sizes/weighting parameters as in MoDL) vs. those with unshared parameters (as in E2E-VarNet). In the former case, it is easy to train with supervision at the output. Even in that case, one can do weak intermediate supervision by training a single cascade first, replicating it for T cascades, and fine-tuning the T-cascade version (as in MoDL). In the latter case, the typical approach is to first train the shared parameter version, then fine-tune the unshared version on that. There are also works that propose intermediate supervision (e.g. doi: 10.3390/e27090929), but the benefit of this is marginal especially in the first shared setup.
- Naturally, the erratic behavior observed is related to the unshared version. This is usually not seen in the shared parameter setup, which has much "flatter" behavior across cascades.

3) Experiments are performed on either DICOM images or single-coil datasets, which have limited utility in MRI.

4) The proposed algorithm only extends to gradient descent/proximal gradient descent type algorithms, and do not explain the more successful variants based on variable splitting (e.g. ADMM)

5) The authors set \sigma = 1, but this has a physical meaning in the derivation as the observation noise, so one cannot arbitrarily set it to any value they want.

6) The authors not only use the flow-based loss term, but multiple other loss terms. It is unclear if the comparison networks used the same additional loss terms. An ablation study on the effect of each term in (13) is clearly missing.

7) 12 steps is not a speed-up over existing unrolled networks. MoDL has 10 steps, E2E-Varnet readily has 12 as well.

Minor points:
- "Φk(·) is the learned regularizer (often implemented by a CNN)"
Often the proximal operator corresponding the regularizer is implemented with a CNN.
- Flow matching literature typically uses the opposite indexing, with x0 being the noise and x1 being the data distribution.
- \sum \delta_k = -1 is an interesting insight, but this is counter-acted by arbitrarily setting \sigma = 1.
- There is something fundamentally wrong with the fully-sampled k-space shown in Fig. 1. Perhaps due to some DICOM processing.

**Questions:**

These are already covered in the weaknesses:
- How does the method work on multi-coil MRI data?
- Why is \sigma = 1? If it is used as the noise level in the dataset, how does it affect the results?
- What is the effect of each term in the loss function?

---

> ### Author Response · Authors · 2025-11-27
> **Response to Reviewer jvCY - part 1**
>
> We thank the reviewer for the constructive feedback. The questions help us improve the justification of our work. Please find our responses to specific queries below.
>
> **Q1: Concerns on the proof.**
>
> **Q1.1: Eq. 1 does not apply to intermediate points on the trajectory. p(y|x_t) would need to be calculated as \int p(y|x_0) p(x_0| x_t) dx_0, since we only know the relationship between y and x_0 (i.e. Eq. 1).**
>
> **A1.1**: Thanks for raising this concern. This is due to a too-strong assumption we made. We can fix the issue in the theory and it does not really affect our algorithm.
>
> We originally assumed the noise, $\epsilon$, is a Gaussian distribution and irrelevant to time step $t$. We then compute the posterior and build the ODE formulation based on this assumption. You are correct that this is not realistic. $\epsilon$ should depend on $t$.
>
> To fix this is not hard. We simply switch from Bayes’ posterior to an energy function, and the proof is not significantly affected. Consequently, there is no impact on the flow ODE formulation, the correspondence between flow ODE and unrolled networks, the algorithm, or the conducted experiments/analysis.
>
> The specific energy function is as follows: $E(x;y)=\frac{1}{2\sigma^2} ||Ax-y||^2 - \log p(x)$. Then, we define a pseudo posterior by setting the negative log-density equal to this energy:  $p_\phi(x|y)\propto \exp(-E_\phi(x;y)$, where $p_\phi(x|y)$ is a pseudo posterior that approximates the true posterior $p(x|y)$, and $p_\phi(x)$ is the learnable prior. Our conditional probability flow ODE is then defined on this energy-based density function: $\frac{dx_t}{dt}=\lambda(t)\nabla_{x}\log p_\phi(x_t|y)$. For this density, we have $\nabla_{x}\log p_\phi(x_t|y)=\nabla_x\log p_\phi(x_t)-A^T(Ax_t-y)/\sigma^2$, which leads to Eq(9). The algorithm is not affected by this change. We have updated our manuscript with this formulation (lines 239 to 254).
>
> **Q1.2: I'm surprised they are trying to come up with a velocity field from k-space to image domain.**
>
> **A1.2** There might be a misunderstanding. We’d like to clarify that all these variables, including $x, y, A, \epsilon$, are in k-space. Though the CNN processes in image space, the $\Phi_k(x_k)$ term (or the $p_\phi(x_t)$ term) takes k-space data as input, converts to image space, transforms using CNN, and converts the output back to k-space. The velocity term is completely computed in k-space.
>
> **Q1.3: The second part of the proof that is questionable is the statement "this velocity aligns with the gradient of the conditional log-density" Why is this true? This is not shown.**
>
> **A1.3**: Regarding alignment between velocity and conditional log-density, we focus on the standard probability flow ODE built upon the energy function, where the velocity is chosen to be the scaled gradient of conditional log density, i.e. we define $v(x_t, t; y)$ as $\lambda(t) \nabla_{x_t} \log p(x_t|y)$. This is widely used in generative models [1], where the score is parameterized by the log density.
>
> [1] Chung, Hyungjin, and Jong Chul Ye. "Score-based diffusion models for accelerated MRI." Medical image analysis 80 (2022): 102479.
>
> **Q1.4 Also fundamentally one would expect (5) to have dependence on y in this setup, instead (5) is describing an unconditional flow on the image set, with no knowledge of measurements.**
>
> **A1.4**: Thanks for pointing out the confusion. It was a typo and $\frac{dx_t}{dt}$ is dependent on $y$. We were already assuming $\frac{dx_t}{dt}$ depends on $y$, as evident in Eq(6,7,8,9). We have fixed the typo in Eq(5) to $\frac{dx_t}{dt}=v(x_t, t;y)$.

---

> ### Author Response · Authors · 2025-11-27
> **Response to Reviewer jvCY - part 2**
>
> **Q2: Concerns on MRI reconstruction literature.**
>
> **Q2.1**: "hyperparameters such as step size and weighting coefficients are typically set through heuristics or empirical tuning" Almost all algorithm unrolling frameworks learn step sizes and weighting coefficients jointly with the proximal operator/regularization neural network. This statement is therefore incorrect. Furthermore, these are considered parameters of the unrolled network, not hyperparameters. This joint learning of regularization and data fidelity parameters is the main advantage of algorithm unrolling over plug-and-play type methods.
>
> **A2.1**: Thanks for pointing this out. We totally agree and indeed many of our baselines and citations (e.g., E2E-VarNet, MoDL) are learning these parameters. We have revised the text accordingly (Line 78). However, such learnable parameters do not really address the issue our flow ODE resolves, i.e., using a flow path to provide intermediate step control and supervision, thus stabilizing the learnt trajectory, and learning efficiency.
>
> **Q2.2**: "they are typically trained with supervision only at the final cascade" This is partially true. First off, it seems the authors are not distinguishing between unrolled networks with shared parameters (i.e. each cascade uses the same CNN and step sizes/weighting parameters as in MoDL) vs. those with unshared parameters (as in E2E-VarNet). In the former case, it is easy to train with supervision at the output. Even in that case, one can do weak intermediate supervision by training a single cascade first, replicating it for T cascades, and fine-tuning the T-cascade version (as in MoDL). In the latter case, the typical approach is to first train the shared parameter version, then fine-tune the unshared version on that. There are also works that propose intermediate supervision (e.g. doi: 10.3390/e27090929), but the benefit of this is marginal especially in the first shared setup.
> Naturally, the erratic behavior observed is related to the unshared version. This is usually not seen in the shared parameter setup, which has much "flatter" behavior across cascades.
>
> **A2.2**: Thanks for pointing out this confusion. What we meant to say is that for existing methods, the supervision information is only derived from the final reconstruction target. This is true even if the network at intermediate cascades share weights (e.g., in MoDL). The final target does not provide sufficient guidance for intermediate steps. On the contrary, our method ensures the intermediate steps to be aligned to the ODE flow, thus achieving superior stability and learning efficiency. We will revise the manuscript to clarify this.
>
> We added additional experiments comparing our method with it on Brainweb dataset and reported the results below. Our FLAT outperforms MoDL as expected in both PSNR and SSIM. Even though MoDL achieves more stability due to the shared parameters, and thus supervision across all steps, it still underperformed due to the limitation of the supervision information, as we explained above. We will add this discussion to our manuscript.
>
> | Method | PSNR | SSIM |
> |---------|--------------|---------------|
> | MoDL|  24.06±2.3748  | 0.7565±0.0900 |
> | Ours   | 33.52±3.3125 | 0.9395±0.0267 |
>
> **Q3: Experiments are on DICOM / single-coil.**
>
> **A3**: We thank the reviewer for pointing out the necessity of conducting experiments on multi-coil data. We have conducted experiments on the fastMRI multi-coil dataset [2], and present the results in the following table. This table compares the performance of the base network and our FLAT in fastMRI multi coil dataset. The Root Sum of Squares (RSS) is applied on multi coil data before computing PSNR and SSIM. The same table has been added in the manuscript in Tab. 7, Appendix H in Page 18. The experimental results indicate that, though the base model (E2E-VarNet) achieves better numerical performance, there is no statistically significant difference between the two results (conducted using a t-test with a 95% confidence interval). This performance is similar to what we observe in fastMRI knee single coil dataset.
>
> | w/ ours | PSNR         | SSIM          |
> |---------|--------------|---------------|
> |         | 29.10±4.1675 | 0.8002±0.1144 |
> | ✓       | 29.02±4.4695 | 0.7913±0.1218 |
>
>
> [2] Knoll, Florian, et al. "fastMRI: A publicly available raw k-space and DICOM dataset of knee images for accelerated MR image reconstruction using machine learning." Radiology: Artificial Intelligence 2020.
>
>
> **Q4: Extending to ADMM.**
>
> **A4**: Thank you for the great suggestion. Our algorithm only applies to gradient descent type algorithms, which cover the majority of works (e.g., E2E-VarNet, MoDL and ISTA-Net). Extending to variable splitting based methods such as ADMM-Net is interesting yet non-trivial. We will consider it as our future work.

---

> ### Author Response · Authors · 2025-11-27
> **Response to Reviewer jvCY - part 3**
>
> **Q5: $\sum \delta_k = -1$ is an interesting insight, but this is counter-acted by arbitrarily setting $\sigma = 1$.**
>
> **A5**:  We thank the reviewer for pointing out that standard deviation $\sigma$ may not be a constant value $1$. We claim that, in an energy-based model, $\sigma$ is a normalized scale factor which works together with $\lambda(t)$ and $\delta$. We have experimented with a variety of $\sigma$ values to study its impact on the final reconstruction results, and show the results in the following table. The same table has been added as Tab. 5 in Appendix D, Page 16. We find that it only very slightly impacts the final result. Hence, setting $\sigma$ to be a constant seems reasonable. Consequently, setting $\sigma=1$ does not weaken the soundness of $\sum \delta_k = -1$.
>
> | $\sigma$ | PSNR         | SSIM          |
> |-------|--------------|---------------|
> | 0.25  | 33.44±3.3331 | 0.9375±0.0281 |
> | 0.5   | 33.29±3.4089 | 0.9359±0.0285 |
> | 1     | 33.52±3.3125 | 0.9395±0.0267 |
> | 2     | 33.44±3.3726 | 0.9368±0.0281 |
>
>
> **Q6:  The authors not only use the flow-based loss term, but multiple other loss terms. It is unclear if the comparison networks used the same additional loss terms. An ablation study on the effect of each term in (13) is clearly missing.**
>
> **A6**: Good question. We present ablation experiments of loss terms in Tab. 1 in the Global Response above (also added as Tab. 4, Appendix D, Page 15 in the revised manuscript). The table demonstrates all loss terms are helpful, although only $L_{velocity}$ is our contribution. The details are discussed in the Global Response.
>
> **Q7: ‘Speed-up’ over unrolled networks.**
>
> **A7**: We claim speed-up over diffusion models such as DDPM/DDIM. Compared to unrolled networks, we do not claim speed up, rather, we claim more stable trajectories.
>
> **Q8: Definition of $\Phi_k(\cdot)$.**
>
> **A8**: We thank the reviewer for the clarification of $\Phi_k(\cdot)$. We have revised the description of $\Phi_k(\cdot)$ to be a proximal regularization block implemented with CNN (see line 203).
>
> **Q9: Indexing notation $x_0$ and $x_1$.**
>
> **A9**: Regarding the indexing, we use $x_0$ to indicate the clean image as we want to be consistent with diffusion model notations.
>
> **Q10: Something wrong with the fully-sampled k-space shown in Fig. 1.**
>
> A10: We are not sure what you mean. Could you please clarify so we can look into it? Thank you.

---

### Official Review · Reviewer_AHYE · 2025-11-03

**Soundness:** 2
**Presentation:** 3
**Contribution:** 2
**Rating:** 2
**Confidence:** 3

**Summary:**

This paper re-interprets unrolling networks as flow matching. It views the iterations derived from optimization algorithms in unrolling networks as time steps in flow matching and aligns intermediate estimation, x_k in unrolling network,with the conditional path x_t in flow matching. Authors formulated it mathematically and claimed the newly derived losses trains unrolling network better. The alignment of optimization iterations in unrolling network to the conditional path in flow matching seems new to me. However the motivation is weak and not well grounded. Please find my concerns below.

**Strengths:**

1. This paper is well-written and has a comprehensive literature review.
2. This paper includes experiments that compares the proposed method to methods that are from different categories.

**Weaknesses:**

1. The authors claim that the core innovation of FLAT defines the velocity alignment between unrolling iterations and flow matching. It is formulated by first defining two different velocities at the k-th timestep:

	a.  **Ideal Discretized Velocity ($v_{t_k}$):** This is the target velocity, defined as the discrete temporal derivative using the ground truth.
	    $v_{t_{k}} = (x_{t_{k}}^{\*} - x^{(k+1)}) / (t_{k} - t_{k+1})$, $x_{t_{k}}^{*}$ is the linearly interpolated ground truth at time $t_k$. $x^{(k+1)}$ is the network's prediction from the *previous* iteration.
	b.  **Network's Predicted Velocity ($v^{(k)}$):** This is the velocity predicted by the network at the current step.
	    $v^{(k)} = (x^{(k)} - x^{(k+1)}) / (t_{k} - t_{k+1})$, $x^{(k)}$ is the network's output at the current step $k$.

        Why is this ideal ode path is better than original path derived from optimization iterations? If this is true, how much influence can this loss terms make?

2. As a following-up questions, how you select the hyper-parameters? Could you explain why there are so many other loss terms? How you balance the weights for each term? Why the weights for the velocity alignment term is so small? Can we only keep the velocity alignment term? What if we have the other loss terms for other unrolling methods.

3. Not sure if it is right to replace $\nabla_{x_t} \log p(x_t)$ with the velocity field $v_\theta(x_t, t)$, as the unrolling network actually does not learn this and has no generative modelling for it. The unrolling network to me just learns a discriminative model. Would you further comment on this?

4. If this velocity alignment help us in training unrolling network, how would the number of discrete steps affect the performance? The more the better?

5. Which specific diffusion model based method you used for comparison? DDPM and DDIM samplers seems to be very broad. Or would you specify how you implement the method using diffusion models?

**Questions:**

Please see weaknesses.

---

> ### Author Response · Authors · 2025-11-27
> **Response to Reviewer AHYE - part 1**
>
> We thank the reviewer for the constructive feedback. The questions help us improve the justification of our work. Please find our responses to specific queries below.
>
> **Q1: Why is this ideal ode path is better than original path derived from optimization iterations? How much influence can this loss terms make?**
>
> **A1**: Our method is the first to take a generative model perspective on unrolled networks. The ideal ODE path acts as a reference path that stabilizes the evolving process and guides important parameters such as step size. In most traditional unrolled networks, the intermediate step parameters such as step sizes are trainable parameters. Although flexible, this also introduces instability. Another issue is these methods only supervise based on information derived from the final reconstruction target, thus there isn’t sufficient guidance for intermediate cascades. Though easy-to-build, these methods lack control especially for intermediate steps, leading to a misaligned, unstable and zig-zag evolving trajectory as shown in Fig. 2 (II).
>
> On the contrary, in our method, the ideal ode path is calculated directly as a target path. Based on this path, we directly interpolate and compute the desired output distribution for every intermediate step. This provides much more powerful supervision information for every intermediate cascade network block. We also determined important parameters such as step size theoretically based on the flow’s geometry. Not surprisingly, this novel theoretical framework brings us a much more stabilized path, and improved reconstruction performance, as shown in Fig. 1(II).
>
> Such comparison can be further illustrated through experiments. In Fig. 4, on Brainweb dataset, we compare our method and traditional unrolled network method (e.g., EVA-VarNet) at every step in terms of their output image quality (measured in PSNR and SSIM). For our method, we observe a stably increasing reconstruction quality through the steps. For E2E-Varnet, however, the output quality oscillates significantly across steps. This is strong evidence highlighting the stability brought by the flow ODE. To further strengthen the message, we have added a similar experiment on fastMRI dataset (Fig. 7, Appendix H, Page 18).
>
> **Q2.1: How do you select the hyper-parameters? Why so many loss terms and how do you balance them?**
>
> **A2.1**: For the hyper parameters, alpha and K control the discretization process of continuous ODE. These two parameters are selected by ablation analysis (see Fig. 6 in Appendix D, Page 15). Weights of the four loss terms are also decided empirically (see Tab.1 in global response). The rationale and necessity of the four loss terms are discussed in the Global Response above. We also have hyper parameters that are used to train the network (i.e. batch size, learning rate, etc). These parameters are inherited from the backbone network (i.e. E2E-VarNet) without changes.
>
> **Q2.2 Why is the weight for the velocity alignment term so small? Can we only keep the velocity alignment term?**
>
> **A2.2**:The weight of the velocity loss term $L_{velocity}$ is small because the loss value is much larger than the other three loss terms. To balance out different loss terms, we have to use a much smaller weight for $L_{velocity}$. The velocity loss term is inversely proportional to the temporal discretization step size ($t_k - t_{k+1}$). Since $t_k = 1-\left(1 - k/K\right)^{(1 + \alpha)}$,  the bigger k is, especially when k is close to K and we are close to the final step, $t_k - t_{k+1}$ gets very small, leading to a large velocity value. To balance the loss value, we use a small coefficient for $L_{velocity}$. The following table shows the values of different loss terms after training ~4k steps on Brainweb dataset. They are very unbalanced. $L_{velocity}$ is at a different magnification compared with other three terms. It will dominate the overall loss if we do not use a very small weight. But using a small weight does not mean this loss is ineffective, as we have demonstrated in the ablation study in the Global response.
>
> | $L_{velocity}$ | $L_{pixel}$ | $L_{perceptual}$ | $L_{semantic}$ |
> |----------------|---------------|-----------------|--------------------|
> |$5.9\times 10^3$          |   0.024      | 0.15            |   0.55              |

---

> ### Author Response · Authors · 2025-11-27
> **Response to Reviewer AHYE - part 2**
>
> **Q2.3 What if we use the other loss terms for other unrolling methods?**
>
> **A2.3**: We admit that there are many types of loss terms that can be used in MRI reconstruction apart from our four losses, such as adversarial loss, total variational loss, cycle consistency loss, etc. In most of the existing unrolled networks, the loss term is typically a combination of the three terms already included in our final loss, such as pixel loss in image space or k-space (like L2 loss or NMSE) and SSIM loss. Very few works use other kinds of loss functions such as feature-level loss [1] and adversarial loss [2]. Utilizing these special losses is interesting yet challenging as they vary from different networks, assumptions and implementations. We are happy to consider it as future work. Having said that, we do not expect adding these other loss terms will diminish the benefit of our flow ODE idea. Our primary loss terms, pixel and perceptual, are the most widely used in unrolled networks. We also used the semantic loss term as a trial to broaden the primary loss.
>
> [1] Wang, Ke, et al. "High fidelity deep learning‐based MRI reconstruction with instance‐wise discriminative feature matching loss." Magnetic Resonance in Medicine 88.1 (2022): 476-491.
>
> [2] Hammernik, Kerstin, et al. "Variational adversarial networks for accelerated MR image reconstruction." Joint Annual Meeting ISMRM-ESMRMB. 2018.
>
> **Q3: Replacing $\nabla_{x_t} \log p(x_t)$ with the velocity field $v_\theta(x_t, t)$? The unrolling network learns a discriminative model, not a generative one.**
>
> **A3**: We need to clarify that replacing the gradient of prior distribution with the velocity field $v_\theta(x_t, t)$ is a standard step in existing flow ODE methods [3]. We do not claim that the unrolled network is a generative model, as it does not learn the full distribution of $p(x_t|y)$. Instead, we claim that the unrolled update rule (Eq. 3) and the discretized flow ODE (Eq. 10) follow an identical mathematical form. The key insight is that the regularization term learned by CNN $\Phi_k(x_k)$ corresponds to the velocity field $v_\theta(x_{t_k}, t_k)$. Therefore, though unrolled networks are discriminative models, they implicitly learn to model the conditional probability flow ODE, and can be better trained from a flow ODE perspective.
>
> [3] Qin, Haina, et al. "Reversing Flow for Image Restoration." CVPR 2025.
>
> **Q4: Number of steps vs performance**
>
> **A4**: We use the parameter $K$ to represent the number of discrete steps in unrolled networks. Ideally a larger $K$ leads to better performance (shown in Fig. 6, Appendix C, Page 15), which is similar to generative models like Diffusion Models and Flow Matching, and vanilla unrolled networks such as E2E-VarNet and MoDL.
>
> **Q5: Which diffusion model architecture was used?**
>
> **A5**: The specific diffusion model we used is MC-DDPM [3], as described in Section 4.1 (Line 423). We trained the model following the original configuration in MC-DDPM, and sampled images using DDPM (1000 steps) and DDIM (12 or 50 steps) sampling strategy.
>
> [3] Xie, Yutong, and Quanzheng Li. "Measurement-conditioned denoising diffusion probabilistic model for under-sampled medical image reconstruction." MICCAI 2022.

---

> > ### Comment · Reviewer_AHYE · 2025-11-27
> >
> > I thank authors for providing this rebuttal. but I am sorry to keep original rating for this work. The table in global response is not significant enough to support whether the velocity term is indeed helpful since the gain is really mininal. this intermediate alignment for training a supervised unrolling network is not well grounded given the ground truth is available, and this is evidenced by the table.

---

> > > ### Author Response · Authors · 2025-11-29
> > > **Response to Reviewer AHYE**
> > >
> > > Thanks for your response and we appreciate your feedback. Our $L_{velocity}$ is useful not only in the metric gains, but also in its robustness. We have conducted ablation experiments for loss terms in Tab. 1 in global response. We selected three types of primary loss functions, and added our $L_{velocity}$ for comparison. The results indicate that no matter which primary loss function we use, our $L_{velocity}$ provides additional performance improvements. This indicates our $L_{velocity}$ a robust loss term, and can be added to any types of primary loss functions.

---

### Author Response · Authors · 2025-11-27
**Global Response**

We thank all the reviewers for their time and insightful feedback. We are encouraged that the reviewers found it exciting and innovative to characterize unrolled networks as a flow ODE (jvCY, nr56, hR2V), and appreciated the comprehensive experiments(AHYE, nr56).

We address each reviewer's specific questions individually. In this global response, we address one common question, i.e., **rationale and efficacy of different loss terms**.

We have 4 loss terms. $L_{pixel}$ and $L_{perceptual}$ are standard fidelity losses in MRI reconstruction tasks, and the $L_{semantic}$ serves as an additional term to provide semantic supervision on MR images (proved to be effective in inverse image problems [1]). In many existing unrolled networks, the loss term is typically a combination of three terms in our final loss, such as pixel loss in image space or k-space (like L2 loss or NMSE) and SSIM loss. Though our FLAT does not require any specific types of loss function as primary loss, we use pixel and perceptual loss in our implementation, because they are the most widely used loss terms in unrolled networks. We also use the semantic loss term as a trial to broaden the primary loss. We propose $L_{velocity}$ as an additional objective to stabilize the iterative process and keep it close to the desired flow, which aligns the ‘velocity’ at each step of unrolled networks to the ODE flow.

Per request of reviewers, we add ablation study to show that all the terms are helpful. See the table below. We have also updated the manuscript with these findings (added to Appendix D Tab. 4 in Page 15). Having done this, we’d like to emphasize that the three reconstruction loss terms are not our technical contribution. The main message is that the flow ODE framework is helpful and the velocity loss term is beneficial.

Table 1. **Ablation of Loss Terms (AHYE, jvCY, hR2V)**. We have conducted the ablation study of loss terms on the Brainweb dataset. **All these four terms contribute to the reconstruction performance, and our $L_{velocity}$ provides additional improvement to the final reconstruction results.**

| $L_{velocity}$ | $L_{perceptual}$ | $L_{pixel}$ | $L_{semantic}$ | PSNR         | SSIM          |
|----------------|------------------|-------------|----------------|--------------|---------------|
|                | ✓                |             |                | 33.06±3.3442 | 0.9361±0.0282 |
| ✓              | ✓                |             |                | 33.62±3.3752 | 0.9412±0.0269 |
|                | ✓                | ✓           |                | 33.15±3.2981 | 0.9350±0.0277 |
| ✓              | ✓                | ✓           |                | 33.44±3.3476 | 0.9378±0.0275 |
|                | ✓                | ✓           | ✓              | 32.83±3.2937 | 0.9325±0.0275 |
| ✓              | ✓                | ✓           | ✓              | 33.52±3.3125 | 0.9395±0.0267 |

[1] Aakerberg, Andreas, et al. "Semantic segmentation guided real-world super-resolution." WACV. 2022.

---

### Meta-Review · Area_Chair_8orf · 2026-01-05

**Summary:**

This paper introduces a framework that reinterprets unrolled reconstruction networks for accelerated MRI reconstruction from under-sampled k-space as conditional probability flow models. It tries to align the optimization iterations in unrolled networks with time steps in flow matching, enforcing intermediate reconstructions consistent with a conditional trajectory rather than only supervising the final output. This perspective imposes trajectory-consistency constraints, and motivates new training losses.

A key strength of this paper is the conceptual contribution in unifying unrolled reconstruction networks with conditional probabilistic flow ODEs, offering a new theoretical framework for understanding unrolled MRI reconstruction. This perspective may provide useful insight for the MRI reconstruction community.

The reviewers have raised several significant concerns. Especially, the theoretical justification and some motivational claims on previous literature are potentially flawed, and it may not convincingly explain why the proposed ideal ODE trajectory should be superior to the original optimization-derived paths compared with previous approaches. The other weaknesses include difficulty to quantify the impact of the complex loss terms especially when compared with other methods, as well as insufficient comparisons to diffusion-based methods.


The author provides a thorough rebuttal with newly added experiments including ablation study table with different loss terms, experiments on multi-coil MR (though not fully reported results), results on different undersampling ratio, and comparison with DDS. I applaud the author’s great efforts to answer reviewers’ questions. These new results do address these questions by providing more details.

After the discussion, one Reviewer AHYE responded to keep score with remaining concerns that the claimed intermediate alignment for training a supervised unrolling network is not well grounded. The other three reviewers didn’t get a chance to attend the discussion. From my point of view, the concerns for Reviewer hR2V and nr56 seems to be mostly addressed, while the response may not fully address the concerns from Reviewer jvCY. Although the author acknowledged several flaws and corrected them in the revised manuscript, the answers to some questions for the proof flaws and incomplete claims may not be fully convincing. While sharing the same remaining concerns with Reviewer AHYE and jvCY, I would also like to add up to the comment from Reviewer nr56 about the comparison with diffusion-based methods: besides DDS, there are many works showing very comparable results for MRI reconstructions with accelerated sampling steps with some based on ODE sampler, which may be very relevant to be considered and compared in this work.

Overall, considering all review comments, rebuttal and discussions, though the paper proposes an interesting concept, at least a major revision with another round of external reviews might be helpful to further strengthen the paper. Thus, I recommend rejection.

**Reviewer Concerns:**

The author provides a thorough rebuttal with newly added experiments including ablation study table with different loss terms, experiments on multi-coil MR (though not fully reported results), results on different undersampling ratio, and comparison with DDS. I applaud the author’s great efforts to answer reviewers’ questions. These new results do address these questions by providing more details.

After the discussion, one Reviewer AHYE responded to keep score with remaining concerns that the claimed intermediate alignment for training a supervised unrolling network is not well grounded. The other three reviewers didn’t get a chance to attend the discussion. From my point of view, the concerns for Reviewer hR2V and nr56 seems to be mostly addressed, while the response may not fully address the concerns from Reviewer jvCY. Although the author acknowledged several flaws and corrected them in the revised manuscript, the answers to some questions for the proof flaws and incomplete claims may not be fully convincing. While sharing the same remaining concerns with Reviewer AHYE and jvCY, I would also like to add up to the comment from Reviewer nr56 about the comparison with diffusion-based methods: besides DDS, there are many works showing very competitive results for MRI reconstructions with accelerated sampling steps with some based on ODE sampler as well, which may be quite relevant to this work and may need to be discussed and compared.

**Reviewer Scores:**

After the discussion, one Reviewer AHYE responded to keep score with remaining concerns that the claimed intermediate alignment for training a supervised unrolling network is not well grounded. Based on the reasoning above, I suspect there may be at least 1-2 reviewers may still lean towards negative scores if they had been able to participate fully in the discussion.

---

### Decision · Program_Chairs · 2026-01-26

Reject